# *Posyandu* Application in Indonesia: From Health Informatics Data Quality Bridging Bottom-Up and Top-Down Policy Implementation

**Afina Faza** [1,2,*], **Fedri Ruluwedrata Rinawan** [3,4,5], **Kuswandewi Mutyara** [3], **Wanda Gusdya Purnama** [6], **Dani Ferdian** [3,4], **Ari Indra Susanti** [3,4], **Didah** [3,4], **Noormarina Indraswari** [3,4] and **Siti Nur Fatimah** [3,4]

1   Master of Public Health Study Program, Faculty of Medicine, Universitas Padjadjaran, Jalan Eyckman, No. 38 Gedung RSP Unpad Lantai 4, Bandung 40161, Indonesia
2   Biomedical Engineering Study Program, School of Electrical Engineering, Telkom University, Jl. Telekomunikasi No. 1, Terusan Buahbatu—Bojongsoang, Sukapura, Dayeuhkolot, Bandung 40257, Indonesia
3   Department of Public Health, Faculty of Medicine, Universitas Padjadjaran, Jalan Ir. Soekarno KM. 21, Jatinangor, Sumedang 45363, Indonesia
4   Center for Health System Study and Health Workforce Education Innovation, Faculty of Medicine, Universitas Padjadjaran, Jl. Eyckman No. 38, Bandung 40161, Indonesia
5   Indonesian Society for Remote Sensing Branch West Java, Gedung 2, Fakultas Perikanan dan Ilmu Kelautan Universitas Padjadjaran. Jl. Ir. Soekarno KM 21, Sumedang 45363, Indonesia
6   Informatics Engineering Study Program, Faculty of Engineering, Universitas Pasundan, Jl. Dr. Setiabudi No. 193, Bandung 40153, Indonesia
*   Correspondence: afina20005@mail.unpad.ac.id

**Abstract:** The community's mother and child health (MCH) and nutrition problems can be overcome through evidence-based health policy. *Posyandu* is an implementation of community empowerment in health promotion strategies. The iPosyandu application (app) is one of the health informatics tools, in which data quality should be considered before any *Posyandu* health interventions are made. This study aims to describe and assess differences in data quality based on the dimensions (completeness, accuracy, and consistency) of the secondary data collected from the app in Purwakarta Regency in 2019–2021. Obstacles and suggestions for improving its implementation were explored. This research applies a mixed-method explanatory approach. Data completeness was identified as the number of reported visits of children under five per year. Data accuracy was analyzed using WHO Z-score anthropometry and implausible Z-score values. Data consistency was measured using Cronbach's alpha coefficient, followed by qualitative research with focus group discussions, in-depth interviews, and field observation notes. The quantitative study results found that some of the data were of good quality. The qualitative research identified the obstacles experienced using the iPosyandu app, one of them being that there were no regulations governing the use of iPosyandu to bridge the needs of the government, and provided suggestions from the field to improve its implementation.

**Keywords:** advocacy; bottom-up approach; data quality; iPosyandu; policy; top-down approach

## 1. Introduction

Currently, the health status in the community is not optimal, partly due to maternal and child health (MCH) and nutritional problems [1–4]. Maternal mortality rates in developing countries are still high and have not reached the targets of the Sustainable Development Goals (SDGs) [1,2]. Infant and under-five mortality rates are still a health problem because of gaps in health services and other factors [1,5]. Another health problem in Indonesia is nutritional problems, including stunting and wasting [2]. These health problems are important because health is one of the human rights and has an essential contribution to the development and progress of a nation [2,6,7].

Health promotion efforts are needed to overcome health problems, consisting of three strategies: advocacy, mediation, and enabling [8,9]. The enabling strategy in the health promotion strategy is implemented through community empowerment in the health sector, which is partly manifested in *Posyandu* (Integrated Service Post) activities. The existence of *Posyandu* in the community is expected to provide convenience in obtaining information and health services for mother and child, monitor the growth of mothers, infants and toddlers, and strengthen nutritional surveillance through data generated from *Posyandu* activities. *Posyandu* activities are implemented voluntarily by *Posyandu* cadres, community members who voluntarily give their time and are able to carry out *Posyandu* activities. Cadres act as extension workers, movers, and liaisons between health workers and the community [10].

*Posyandu* has five activities (as shown in Figure 1), starting from registration of the children and mothers by the cadre, then measurement of body height and weight of children under-five and mothers. After that, the cadre records the data into the mother and child's MCH book, as well as their registration book. The next step is health education by cadres or health workers from *Puskesmas* (community health care centers), then health services from health workers, e.g., immunization, contraception, antenatal care, and postnatal care [6].

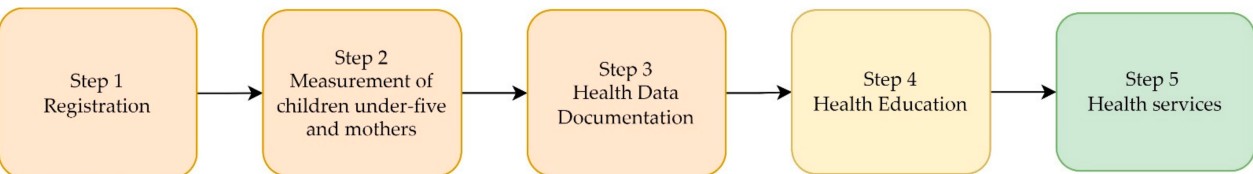

**Figure 1.** Posyandu activities.

In the current digital era of Web 3.0 (with the involvement of artificial intelligence and the internet of things), vast opportunities have opened up for the public to easily access and obtain health information [11,12]. Health service providers can utilize this opportunity to promote health in a digital context [12,13]. One of the implementation efforts of health promotion innovations in community-based health is the digitalization recording and reporting of *Posyandu* activities by cadres using the iPosyandu application [14,15]. iPosyandu is a health informatics tool (HIT) registered in Google Play in December 2018, which serves to collect, store, and process data obtained from *Posyandu* activities [14–16].

Based on the picture above, Figure 2a shows the homepage of the iPosyandu app, and then Figure 2b shows the login screen for the iPosyandu application. When the cadre uses it, after the cadre registers an account, logging in proceeds by using a cellphone number and password. Figure 2c displays when the page that the cadre will use to enter data on the results of *Posyandu* activities. The data for infants and toddlers includes information on the identity of infants and toddlers, the identity of the parents, the results of measurements of weight and height, immunizations, and the administration of vitamin A. Then, the data are stored in the iPosyandu database.

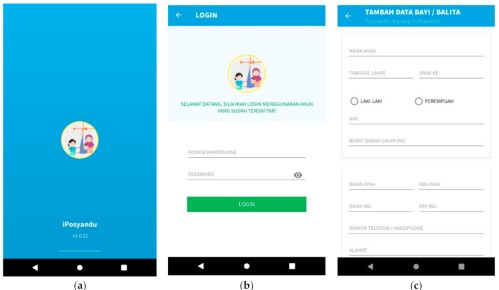

**Figure 2.** Graphical interface of iPosyandu: (**a**) homepage, (**b**) login page, (**c**) children's register page.

Data problems are essential in the health information system because health data is intended for various things, so good data quality is needed [17]. Good data quality is necessary for advocacy steps and supports evidence-based health policies. Hence, the data collected becomes meaningful and material for determining the next policy direction [18–20]. Based on the above conditions, assessing the data's quality is necessary. It is hoped that good data quality can produce innovation in the form of good data synchronization to bridge and facilitate the needs of the community and government. Advocacy efforts are needed to increase its usefulness [8]. The targets in health advocacy are various parties who contribute and can provide support for implementing the programs. Decision makers and policymakers in government institutions are the targets of advocacy, whose role is to formulate health problems in the community, determine issues that require advocacy efforts, and then determine policy directions. The existence of data quality plays a role in the first advocacy step by providing credible and appropriate quality data evidence [21,22]. The following advocacy step presents the relevance of the issues discussed with policies in the health sector and across sectors, then drafts an outline of problem solutions and follow-up plans, followed by determination of the right advocacy strategy [22]. It is expected that advocacy steps can produce policy. Thus, implementation needs an approach.

## 2. Materials and Methods

The design in this study was a sequential explanatory mixed method, starting with quantitative and continuing with qualitative research [23]. In the quantitative research, the post-positivism paradigm is used; then, in qualitative research, the interpretivism paradigm with a case study approach is used [23,24]. In the quantitative part, cross-sectional design was used by analyzing secondary data from the iPosyandu database that was processed systematically and objectively to assess data quality [23]. The study area was in Purwakarta Regency, West Java, Indonesia, as the pilot project location. We observed the data quality for the last three years (2019–2021). Data completeness was analyzed regarding how complete the data reported by *Posyandu* was, according to the reporting period. *Posyandu* activities are carried out regularly, ideally once a month; as such, in 1 year, it is expected that there will be 12 visits. In this research, we identified data completeness using the number of children's visit, which has three categories: Not complete (1–4 visits reported per year); Less complete (5–8 visits reported per year); and Complete (9–12 visits reported per year) [15]. The accuracy of the input data includes the accurate value of age and measurement data (in kilograms and centimeters). Data accuracy is assessed based on the WHO Z-score Child Growth Standards by having an implausible value (a value outside the normal range) of WHO Z-score: weight per age, height per age, weight per height (waz, haz, whz) of $>5$ standard deviations (SD) and $<-6$ SD, $>6$ SD and $<-6$ SD, $>5$ SD and $<-5$ SD, respectively [15]. Afterward, we identified the proportion of data completeness and accuracy. The chi-square test was used to analyze the difference in completeness and accuracy over 3 years. Consistency is defined by the coherence of the same data element at different times. Consistency assesses the presence of outliers regarding the possession of MCH books, immunizations and annual vitamin A administration, and supplementary feeding, then analyzed using Cronbach alpha. STATA version 15.1 Special Edition License (StataCorp LLC, College Station, TX 77845, USA) was used for the analysis.

The qualitative research was conducted at the location using focus group discussion (FGD) with cadres and midwives (in villages and *Puskesmas*). Then, in-depth interviews with nutritionists, the Head of Family Health and Nutrition Division from the Health Office, and field notes observation were conducted. All participants were interviewed with open questions about obstacles and suggestions for improving the implementation of recording and reporting activities using iPosyandu. A qualitative sample was chosen using purposive sampling based on the following criteria: (1) Cadre who has officially become active in *Posyandu* activities for the last three months and uses the iPosyandu application; (2) Midwife who serves in the village and has education in the Midwifery field; (3) Midwife who serves as a coordinator at *Puskesmas* and has education in the Midwifery field;

(4) Nutritionist who serves at *Puskesmas* and the has education in the Nutrition field; (5) Head of Family Health and Nutrition Division from the Health Office who serves as part of the Mother and Child Health program management. Table 1 shows the role of participants in the qualitative research and the total subject involved.

**Table 1.** The role of participants in qualitative research.

| Participant | Role | n |
|---|---|---|
| Cadre | Users of the iPosyandu application and input the data from *Posyandu* activities | 28 |
| Village midwife | Cadre's supervisors in each village | 14 |
| Coordinator midwife | Cadre's supervisors in each *Puskesmas* | 2 |
| Nutritionist | Monitoring data from *Posyandu* activities | 2 |
| Head of Family Health and Nutrition Division of the Health Office | Providing information and considerations regarding the follow-up plan for the iPosyandu application in the advocacy step | 1 |
| | Total | 47 |

The qualitative data were collected, and the analysis process started with transcription. Data from qualitative research in the form of recordings were transcribed in written form or related to information from the results of focus group discussion activities, in-depth interviews, and field notes. The transcribed recordings were then transferred to a computer using NVivo Release 1.6.1 software QSR International (Burlington, MA, USA). After that, we carried on the condensation process to sort and screen data based on the transcription results considered necessary for analysis. The third step was the coding process to break down data into smaller units: words, groups of words, paragraphs, or parts of data that have meaning. The fourth step was the categorization process, grouping the coding results that have similarities, with one category consisting of several codes, and then they are labeled. The fifth step was to describe the theme, identify the relationship between codes and categories in the data, and then develop conclusions with concepts and narrative text. The following process was to verify the truth and trustworthiness of transcription, reduction, coding, and categorization stages. The last step was the presentation of research data analysis results that are interesting and representative.

### 3. Results

*3.1. Quantitative Result*

The descriptive analysis in this study showed that 3 out of 20 community health care centers (*Puskesmas*) in Purwakarta were actively filling out iPosyandu data. Table 2 shows the amount of data for children under-five from iPosyandu. The highest one was *Puskesmas* Pasawahan, as the location of the pilot project, with 17,168 data inputs (94.84%), followed by *Puskesmas* Koncara, with 748 data inputs (4.13%), and *Puskesmas* Jatiluhur with 186 data inputs (1.03%). This condition occured because not all *Puskesmas* have received direct dissemination of information and training due to limited face-to-face meetings during the pandemic. Still, webinars and online training activities regarding iPosyandu werecarried out through cadres, midwives, and nutritionist representatives from each *Puskesmas* in Purwakarta Regency.

Each village consists of several *Posyandu*. The village that inputted the highest number of data during the years 2019–2021 was Pasawahan Kidul village with 3449 data inputs (19.05%), followed by Pasawahan village with 3382 data inputs (18.68%), then Cihuni village with 2570 data inputs (14.2%). All of them were located in the working area of the *Puskesmas* Pasawahan (as shown in Figure 3). This condition occurred because *Puskesmas* Pasawahan has received direct training and regular monitoring regarding data input.

**Table 2.** The amount of data for children under-five from iPosyandu.

| *Puskesmas* (Community Health Care Center) | Frequency | Percentage (%) |
|---|---|---|
| *Puskesmas* Pasawahan | 17,168 | 94.84 |
| *Puskesmas* Koncara | 748 | 4.13 |
| *Puskesmas* Jatiluhur | 186 | 1.03 |
| Total | 18,102 | 100 |

**Figure 3.** Data input percentage in each village.

The number of *Posyandu* currently registered and actively inputting the data is 69 out of 148 *Posyandu* from three *Puskesmas*. The *Posyandu* that inputs the highest number of data was *Posyandu* Melati III, located in Cihuni Village, with 1929 data inputs (10.66%), followed by *Posyandu* Manggis II with 1655 data inputs (9.14%), located in Pasawahan Kidul Village (as shown in Figure 4). The two *Posyandu* are located at the *Puskesmas* Pasawahan, which has been monitored and received direct training from the start.

Based on the description above, the amount of incoming data has shown excellent initiative results. However, there are still gaps in the number of users and incoming data, so further identification of the obstacles experienced in the implementation of recording and reporting activities through the iPosyandu application is needed.

Based on the analysis of the data completeness, there has been a change in the total number of visits. In 2019, the visits reached 10,551 visits, and then in 2020, the visits decreased to 2605 visits; however, in 2021, the number of visits began to increase again by 3570 visits (as shown in Table 3).

Data Percentage each Posyandu

**Figure 4.** Data input percentage in each *Posyandu*.

**Table 3.** Data completeness.

| Category | Year | | | | | | Year | % | *p*-Value |
|---|---|---|---|---|---|---|---|---|---|
| | **2019** | **%** | **2020** | **%** | **2021** | **%** | | | |
| 9–12 Report | 4080 | (38.67%) | 764 | (29.33%) | 1576 | (44.15%) | 6420 | (38.38%) | |
| 5–8 Report | 3726 | (35.31%) | 455 | (17.47%) | 1032 | (28.91%) | 5213 | (31.17%) | 0.000 |
| 1–4 Report | 2745 | (26.02%) | 1386 | (53.21%) | 962 | (26.95%) | 5093 | (30.45%) | |
| Total | 10,551 | (100%) | 2605 | (100%) | 3507 | (100%) | 16,726 | (100%) | |

The table above shows that the iPosyandu data observed for three years showed a good start. In 2019, the complete data was 38.67%, then in 2020, it decreased to 29.33%, and then, in 2021, it began to increase again to 44.15%. The declining number of visits was due to various conditions in the field, including the COVID-19 pandemic that occurred at the end of 2019 and is still ongoing in 2022, causing *Posyandu* activities to be hampered and even stopped, resulting in a decrease in the number of *Posyandu* visits and reports. This declining condition was also due to limited direct communication between cadres who input data and midwives who act as *Posyandu* service supervisors in the field. Training activities that were limited to only cadre representatives were also affected because the information dissemination process was limited, so only cadre representatives could input data into iPosyandu. Based on the chi-square test results, it was found that a *p*-value of 0.000 indicated a significant difference in data completeness.

The dimension of data accuracy shows a good initiative start, but then decreased in 2020 because of several conditions, such as the COVID-19 pandemic, and increased in the following year; the outliers continuously decreased in these three years.

Based on the analysis results in Table 4, it was found that accurate data in 2019 reached 9523 inputs, or around 86.88%, and outliers made up 1438 inputs, or 13.12%. In 2020, accurate data amounted to 1508 inputs, and thus experienced a significant decrease. One of the reasons for this condition is the pandemic; the amount of incoming data decreased because the *Posyandu* activities were postponed. Limited communication during the pandemic directly causes training in the recording and reporting process and medical

and technical training regarding growth measurement to be hampered. Supervision from the *Puskesmas* through midwives in *Posyandu* activities also experienced limitations. In 2021, the amount of accurate data increased by 3247 inputs, or about 89.82%, with data outliers experiencing a significant decrease, at only 368 data inputs, or approximately 10.18%. Based on the results of the chi-square test, it was found that the *p*-value of 0.000 indicates that there is a significant difference in data accuracy. Overall, the accurate data shows a reasonably good result, as noted in the number of data outliers that have decreased significantly.

**Table 4.** Data accuracy.

| Data Accuracy | Year | | | | | | Total | % | *p*- Value |
|---|---|---|---|---|---|---|---|---|---|
| | 2019 | % | 2020 | % | 2021 | % | | | |
| Accurate | 9.523 | (86.88%) | 1.508 | (53.70%) | 3.247 | (89.82%) | 14.278 | (82.13%) | 0.000 |
| Outliers | 1.438 | (13.12%) | 1.300 | (46.30%) | 368 | (10.18%) | 3.106 | (17.87%) | |
| Total | 10.961 | (100%) | 2.808 | (100%) | 3.615 | (100%) | 17.384 | (100%) | |

Data consistency in 2019 showed a scale reliability coefficient of 0.2408, which can be influenced by various factors, including considering that this application has just been used. In 2020, there was an increase, with the scale reliability coefficient becoming 0.4457. In 2021, the consistency decreased to 0.2767 (as shown in Figure 5).

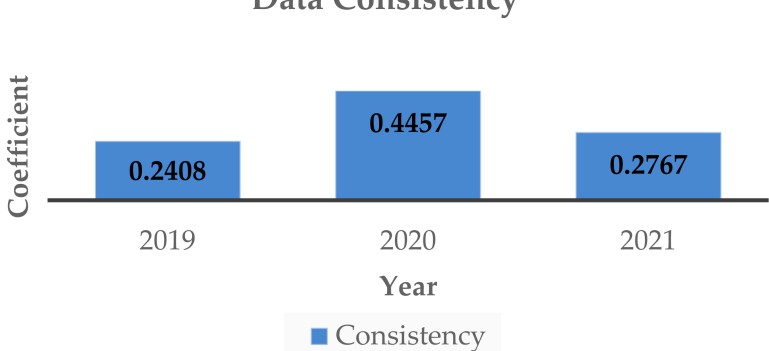

**Figure 5.** Data consistency.

Overall, the quantitative research results in this study are that some of the data were of good quality. The results showed that the amount of data entered at the beginning was reasonable, then decreased due to the pandemic and other obstacles. In the following year, it began to show an increase. The level of data accuracy also increased, indicated by decreasing data outliers. This result requires further identification of the obstacles experienced in recording and reporting activities through the iPosyandu application, suggestions, and further improvements.

We hypothesized that there are differences in data quality (completeness, accuracy, and consistency) of iPosyandu between 2019, 2020, and 2021. The chi-square test results both on the dimension of data completeness and accuracy obtained significant *p*-value (Tables 3 and 4), while the consistency was <0.70 (0.2408; 0.4457; 0.2767).

### 3.2. Qualitative Result

The result of qualitative research is the barrier to implementing iPosyandu, which consists of several factors such as the absence of regulations for using iPosyandu, human resources, pandemics, and connection with the front-end of the government's applications, facilities, and infrastructure (see Figure 6).

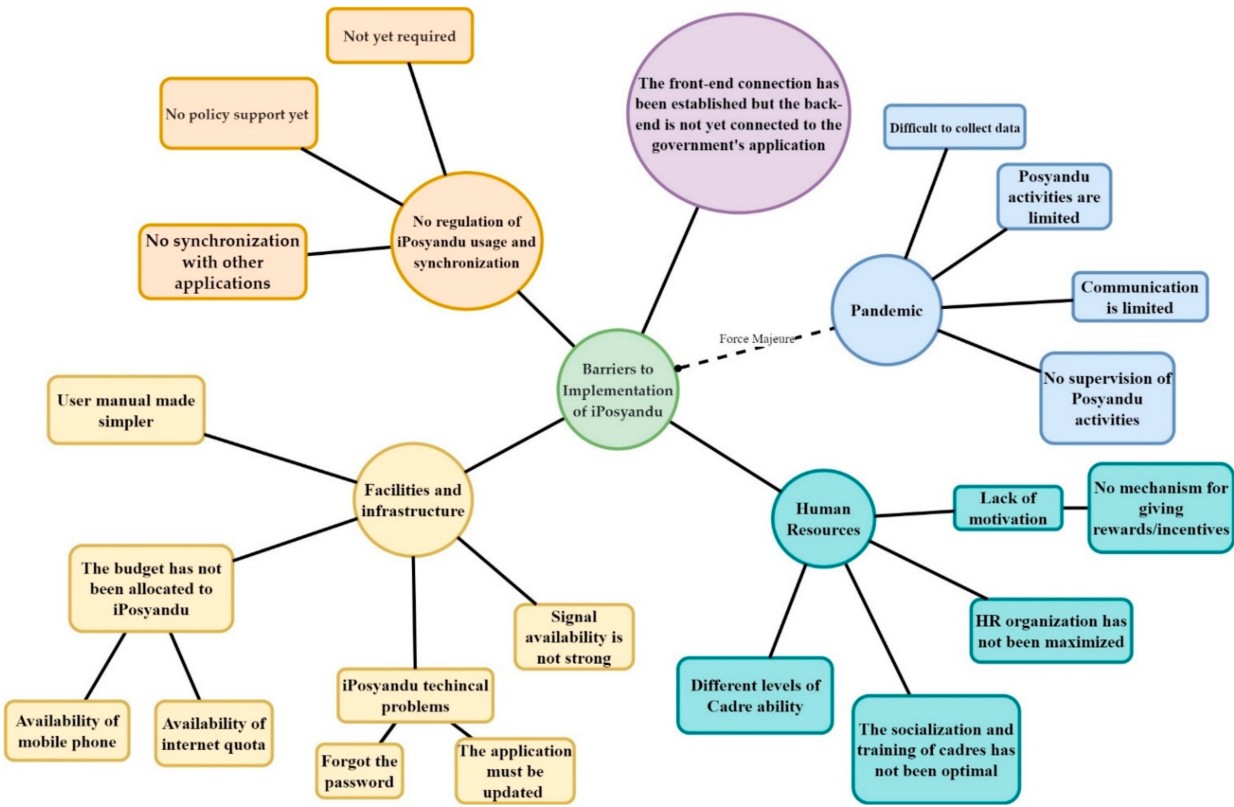

**Figure 6.** The barrier to the implementation of iPosyandu.

### 3.2.1. Regulation

The absence of regulations has become a crucial issue in the implementation of recording and reporting activities using iPosyandu. Officers from the *Puskesmas* who act as supervisors in the field cannot require cadres to fill in the iPosyandu because there are no regulations that require the use of iPosyandu regularly. With the presence of instructions, it is also hoped that the relevant stakeholders can collaborate. The informant stated that:

> *"For example, if there are instructions from all, for example, all Posyandu must use the iPosyandu, for example, it must be filled in, say, in the application, it seems like it will, and the village from the sub-district already knows so that they will facilitate it."*

The description above illustrates how important a policy is in creating a change at a higher level. The existence of instructions from related offices, such as the Health Office, the Community and Village Empowerment Office, and related stakeholders, can collaborate to overcome obstacles to the use of iPosyandu, i.e., through the provision of facilities that support recording and reporting using iPosyandu.

### 3.2.2. Human Resources

The second obstacle is regarding human resources (HR). The implementation of iPosyandu as a new health informatic tool is faced with conditions where users have different levels of ability in operating Android devices. This is a challenge because the application that has been built properly will not work if the user cannot use it. Maximum benefits can be achieved by following the objectives. Different levels of ability are also exacerbated by the condition of technological resistance experienced by users. Another condition is the age factor of cadres, who have various age ranges. Older cadres experience more significant difficulties than younger cadres when operating Android devices and using iPosyandu. During the FGD with the midwife, the informant stated:

> *"There are some cadres who can't operate the application."*

This statement shows that the application's obstacles are that not all cadres can operate the app for recording and reporting activities using the iPosyandu application. Moreover, during the cadre FGD, the informant said that:

*"if we just alone, there's no one can correct."*

The cadre said that the dissemination of information was limited to representatives of cadres in each *Posyandu*, causing cadres to be unable to correct each other if there were errors or difficulties in using the iPosyandu application. Dissemination of information was also hampered due to the pandemic, so this activity was carried out online through the Zoom meeting platform. However, this also encountered obstacles because the information conveyed to cadres felt less than optimal. The informant stated this in the Cadre FGD:

*"Zoom meeting, I'm only present, and listen."*

Based on these respondent's descriptions, online training seemed less effective because cadres only attended and listened, and it was not easy to practice directly with a facilitator.

The discussion on HR cannot be separated from the importance of organizing HR in the continuity of recording and reporting activities using iPosyandu. The current condition of the workload of cadres is significant and concurrent in several programs; however, the number of cadres is minimal. This situation has contributed to the impediment of the implementation of iPosyandu. In the cadre FGD activity, the informant revealed:

*"a lot of work. The cadre is the same person."*

Currently, cadres are not only focused on the main activities of *Posyandu*, but are also involved in other programs such as screening for *Pendidikan Anak Usia Dini* (Early Childhood Education Programs) and *Taman Kanak-Kanak* (Kindergarten), as well as programs related to the elderly. This means that cadres may be unable to focus on inputting data into the iPosyandu app. Another thing that plays a role is that there is no division of tasks to input data to iPosyandu. This condition is described in the cadre FGD, and the informant said that:

*"Tell anyone to input, but they don't want to do that, so we're alone."*

Currently, not all cadres have an active role in inputting data to iPosyandu due to the absence of a clear division of tasks or determining their turn to input data. In the end, the data input process was only carried out by one or two cadres, especially cadres representatives who had received the training. Other factors that affect a cadre's motivation is a lack of enthusiasm in inputting data and because *Posyandu* activities are voluntary community empowerment activities carried out by cadres. There is no mechanism for providing rewards/incentives for cadres who input data into iPosyandu.

### 3.2.3. Force Majeure (Pandemic)

The next obstacle experienced was force majeure, such as pandemic conditions. This condition resulted in limited *Posyandu* activities, and even halted activities completely, making data difficult to obtain. This was stated by an informant at the cadre FGD that:

*"Yes, because the Posyandu is also twice a year for the past two years."*

*Posyandu* activities should ideally be held once a month, but due to the pandemic conditions, *Posyandu* only runs twice a year. This has also led to a decrease in the number of *Posyandu* visits. The pandemic also causes communication between health workers at the *Puskesmas* and cadres to be hampered. Health workers cannot directly supervise *Posyandu* services, such as measuring the growth of infants and toddlers, and cannot provide health services, such as contraceptive and immunization services at *Posyandu*.

### 3.2.4. Infrastructure

Infrastructure has several obstacles to implementing iPosyandu recording and reporting activities. The availability of mobile phones, internet quota, and signals are topics that are widely discussed in FGD activities and in-depth interviews because iPosyandu is one of the health informatics tools based on mobile applications and websites. This makes the

availability of mobile phones, signal quality, and internet quota important in using iPosyandu. Without a cellphone, internet connection, and a strong signal, the iPosyandu application cannot be utilized optimally. Users, namely cadres, experienced obstacles in the availability of these facilities and infrastructure, as stated by an informant at the cadre FGD:

> *"I don't have an Android."*

Some cadres do not have android phones, or the phone cannot be used, or have limited cellphone capacity. Most cadres also input iPosyandu data using personal cellphones, so cellphones are also used by the children of the cadres for online school activities during the pandemic. This is also an obstacle in recording and reporting activities using iPosyandu. The barrier related to the availability of internet signal and quota was described by the informant in the cadre FGD:

> *"If you use iPosyandu, it feels good. It's just that sometimes the problem is the quota, and sometimes there's no signal."*

Funds for internet quotas still require personal funds from cadres, so the quota is limited for recording and reporting using iPosyandu. In several areas in the research location, namely in Purwakarta Regency, some areas still experience signal difficulties. Currently, the iPosyandu application requires an internet connection to operate, so internet quota and signal constraints are crucial. The provision of facilities and infrastructure, such as mobile phones and internet quotas, is a large drain on the operational funds for *Posyandu* activities.

### 3.2.5. Connection with the Front-End of the Government's Applications

The next obstacle experienced was because iPosyandu had not been connected to the government's app on the back-end. Instead, it was connected on the front-end, so recording and reporting had to be done repeatedly. This situation was stated by the informant in the midwife FGD activity that:

> *"So it means that even at iPosyandu, we still enter e-PPGBM, it's different."*

The data from *Posyandu* activities that have been recorded in iPosyandu need to be inputted into the *e-PPGBM/elektronik-Pencatatan dan Pelaporan Gizi Berbasis masyarakat* (electronic community-based nutrition recording and reporting) by the officers at the *Puskesmas*. This is because the back-end of the iPosyandu application has not yet been connected to other government's back-end applications, one of which is *e-PPGBM.*

Throughout the use of iPosyandu from 2017 to 2022, there have been no regulations governing the use of iPosyandu and synchronization. Using the iPosyandu application is still not mandatory, so users have not focused and endeavored to fill out iPosyandu regularly. This was stated in the cadre FGD activities. The informant said that:

> *"We're not obligated yet. That's why we don't fill it out, not for other reasons."*

In qualitative research, we also discover suggestions for improvement. We classify them into bottom-up, top-down, and bridging approaches (see Figure 7). The bottom-up section requires an improvement from human resources, which includes the division of data input tasks to iPosyandu, and a qualified cadre to input data to iPosyandu. The division of data input tasks to iPosyandu can take turns managing the duties and responsibilities of each cadre so that recording and reporting can be carried out routinely. This includes the division of data input tasks to iPosyandu, then special officers to input data to iPosyandu, as described in the cadre FGD activity. The informant said:

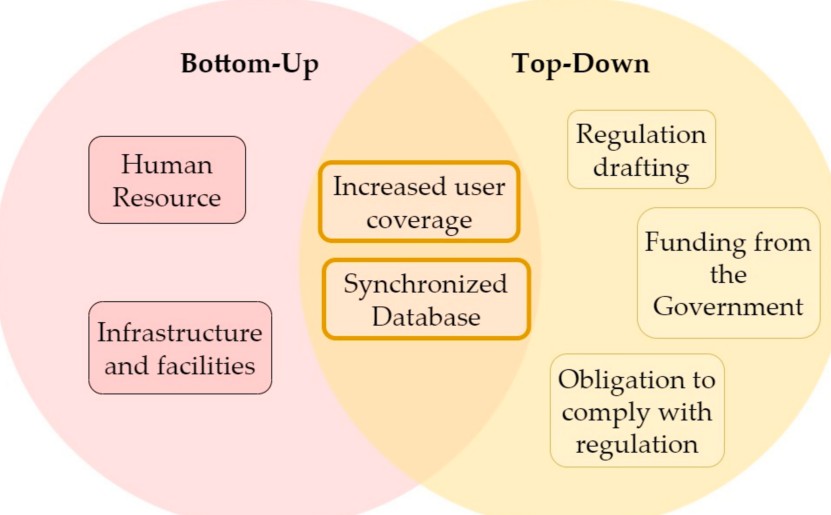

**Figure 7.** Bridging approaches between bottom-up and top-down policy implementation.

> *"(The fill of) iPosyandu is the turn if we are like that, so don't just one, be backed up."*

The division of data input tasks to iPosyandu can take turns managing the duties and responsibilities of each cadre so that recording and reporting can be carried out routinely. The next improvement suggestion is to collaborate with cadres from other programs. Another suggestion is a training program for all cadres to increase their motivation. In the in-depth interview with the nutrition officer, the informant said:

> *"I really want it... so yesterday it was more to the head of the Posyandu who was invited here to be given information and training. So if I want all members of the Posyandu, if possible, for example, one, then several posts, but all members join. So let's all do it, so let's help each other, right?"*

It is hoped that the dissemination of information and training activities that involve all cadres can increase the understanding of cadres about iPosyandu, as well as encourage helping each other in the early stages when problems occur in the field, to be forwarded to the iPosyandu administrator. This is supported by statements from informants in the cadre FGD activities:

> *"Training is possible, but not only for one person. It's better if all cadres. The old ones, who don't have androids, join all iPosyandu training because iPosyandu is not only one Cadre who must know, but all cadres must be able to work together."*

The description conveys the spirit of community empowerment, with the principle of collaboration beginning with cooperation between cadres in *Posyandu*. All cadres, regardless of age, cellphone ownership, or other challenging conditions, must be involved in the dissemination of information and also training activities because the purpose and benefits of iPosyandu can be obtained through the participation of all cadres.

The next suggestion is regarding the provision of facilities and infrastructure, such as specific mobile phone facilities for recording and reporting using iPosyandu. This is mentioned in the cadre FGD activities. The informant said:

> *"The main thing is to facilitate the cellphone, ma'am, so it doesn't mix with ours, so it doesn't mix with WA, telephone, specifically for applications like that."*

The provision of special mobile phone facilities for the iPosyandu application aims to prevent mobile phones from being integrated with personal needs so that the application load on mobile phones is not too heavy and there is adequate storage space so that the recording and reporting process through iPosyandu can run optimally.

Suggestions for further improvement are related to the provision of internet quota facilities by the village funds. This was stated by the informant in the cadre FGD activity that:

*"The point is the quota, now there's a cellphone too, if there's no quota too, now iPosyandu has to (be filled)"*

The need for internet quota availability is one of the essential things in implementing recording and reporting through iPosyandu. The application cannot be used without an internet quota, so suggestions for improvement regarding the provision of internet quota facilities are a priority in the discussion of suggestions for improvement.

Furthermore, it is also hoped that the database can be synchronized, such as being connected to government-owned applications through the back-end, and with synchronization to various other applications. In the in-depth interview with the nutritionist, the informant said:

*"Helpful with input process to.. especially if it's linked with e-PPGBM, that's really more... more helpful, really more helpful."*

Indeed, iPosyandu synchronization with the government app on the back-end is expected to make it easier to report the results of *Posyandu* activities. The data generated are real-time and can shorten the reporting process time. Along with the development of time, increasing the coverage of iPosyandu users through increasing the number of iPosyandu users needs to be pursued so that the perceived benefits can reach more *Posyandu*. Another suggestion for improvement is to coach each *Posyandu*. In the in-depth interview with the nutritionist, the informant stated that:

*"If we see conditions like this in our place, it can't be done immediately in a year. It can be like that, you know. So we have to go slowly, so we have to be one, one Posyandu first, so it's fostered"*

The technology dissemination process in the community requires an intensive mentoring process so cadres can achieve independence in recording and reporting activities using the iPosyandu application. Regulation of the use of iPosyandu is needed for change not only at the *Posyandu* level, but also a higher level. The existence of regulations also serves as the basis for allocating funds and the obligation to use iPosyandu, which goes hand in hand with gathering support from stakeholders. The informant explained this in the midwife FGD activity:

*"So if for example there are instructions from above, for example the Health Service wants it like this, but we don't facilitate it, but they because the budget is passed down through the village, so the village will facilitate it like we give it, for example maybe later from the tea village budget to buy the mobile phone"*

Raising support from relevant stakeholders such as the Regency Health Office and the Community and Village Empowerment Office can support the formation of regulations for the use of iPosyandu. Funding for *Posyandu* activities comes from various sources, one of which being village funds, so it is hoped that the regulations governing the use of iPosyandu can be the basis for submitting facilities that support the recording and reporting activities using iPosyandu.

## 4. Discussion

The use of the iPosyandu app and website is expected to be a bridge between the community and the government. However, this can only be effective and efficient when the quality of the data generated from the iPosyandu application and the website is good. In our three years of initial research, over than half of the data were of good quality. This result is in line with previous research, which stated that the quality of the iPosyandu data in Indonesia was partially good [15]. The quality of the data, some of which were of good quality, was related to previous research, which suggested that the iPosyandu application was easy to use [25]. Other studies stated that various things affect data quality

in health data, such as managerial support, resources, regulatory capabilities, information technology alignment, staff participation, and data/system integration [26].

Policy implementation through bottom-up and top-down approaches is needed [27]. This requires bridging the bottom-up and top-down approaches for effective implementation, as shown in Figure 7. According to the author's point of view, the bridging approaches can complete the implementation between bottom-up and top-down approaches [27], which previous recent reviews found, e.g., in the environmental field [28] and healthcare [29]. Currently, the WHO states that, generally, both approaches complete each other; however, regarding health informatics tools, the national system is used, but community mobile-health is not discussed [30].

From Figure 7, the barriers to human resources can be overcome by organizing them. Barriers in HR can also affect user motivation, impacting the implementation of iPosyandu reporting recording activities that affect data quality [31]. In previous research, users lost motivation to continue using the application [32]. One approach to increasing motivation is the provision of rewards or incentives to cadres as a form of appreciation and increasing motivation for cadres to input data regularly, completely, accurately, and consistently. Previous research has stated that an incentive mechanism can increase user motivation, significantly impacting data quality [31]. The WHO also stated that providing incentives for cadres directly impacts the effectiveness and sustainability of health programs and will increase cadre motivation [33].

Technology acceptance also affects implementing the iPosyandu app, which functions as a reporting bridge from what was previously manual to electronic. This obstacle also occurs in other studies, which describe barriers to technology acceptance during the transition to the use of health informatics tools. Previous research on iPosyandu suggests that the quality of the iPosyandu application will affect the level of satisfaction of cadres as users, which will have an impact on the use of the iPosyandu application [25].

This study also found that the dissemination of information and training was limited to only a few cadres, which would also affect the implementation of iPosyandu recording and reporting activities. Previous research has stated that limited training will create barriers to the use of technology [34]. Other studies say that adequate and appropriate training is an important thing that affects data [26]. The cadre's ability level as a user and in inputting data will also affect the data quality. This is in line with previous research stating that there is a relationship between one's ability and the quality of the data produced [26,35,36].

From top-down approaches as shown in Figure 7, government regulations and support from stakeholders also play an important role. The process of making a policy is complex and dynamic. Many obstacles are experienced, and support from the government and related stakeholders is essential in supporting changes at a higher level environment, and broader level, in this case, related to changes in manual reporting to digital through the iPosyandu application in Purwakarta Regency. Collaboration is needed in advocacy efforts with the Ministry of Health, the Health Office, the Community and Village Empowerment Service, Regional Government, and related stakeholders. This is also in line with previous research, which requires a strategy and collaboration of the Central Government, Regional Government, health service systems, cadres, and academics to support a policy [37–39]. Expected support can take the form of a circular or decree, and may even take the form of local regulation. This support is expected to be comprehensive, from the government and stakeholder level, down to the grassroots level. Coordination across regional apparatus organizations involving the Health Office (*Dinas Kesehatan*), the Community and Village Empowerment Service *(Dinas Pemberdayaan Masyarakat dan Desa)*, and other stakeholders is needed when implementing a policy. This coordination is expected to overcome obstacles in the field and expand the scope of using iPosyandu in the Purwakarta Regency to minimize the risk of reporting delays and other conditions caused by manual reporting. Previous research stated that the participation and cooperation of all staff would affect the quality of health data [26,40].

Previous research on data quality has been carried out, but has not yet reached the advocacy step [15]. Advocacy steps must be pursued to overcome obstacles in recording and reporting activities through iPosyandu. The government is directly responsible for funding, implementing, and operating health informatic tools infrastructure [38]. Decentralization from the central government to the local government, such as in the Purwakarta Regency, can increase flexibility in funding allocation for implementing decisions and policies [41]. One form of implementation of central government decentralization is village funds. Village funds are funds sourced from the State Revenue and Expenditure Budget *(Anggaran Pendapatan dan Belanja Negara)* provided through the district/city Regional Revenue and Expenditure Budget *(Anggaran Pendapatan dan Belanja Daerah)* to be used in community development activities and community empowerment [42]. The budget from village funding *(Dana Desa)* is expected to be allocated to provide facilities that support reporting using iPosyandu. It can provide infrastructure such as gadgets (mobile phones), internet quotas, strengthening telecommunication signals, and human resources in the form of honorariums or rewards. Another source of funds that can be used is grants. The West Java Provincial Government has allocated funds for *Posyandu* and *Posyandu* cadres [43]. In Governor Regulation Number 66 of 2020, Article 7 describes that grants are used for the operation of Posyandu activities and the processes of Posyandu's cadres [44]. It is hoped that various sources of funding for Posyandu can support the implementation of recording and reporting activities using iPosyandu.

In this research, we found that feedback from cadres, midwives, and nutritionists (bottom-up) coincides with feedback from the Regency Governmental Officers (top-down), which is shown as the bridging part between both approaches. Increasing user coverage by massive dissemination of information and training is essential. The process of synchronizing the database is also necessary to improve the benefit and effectiveness of iPosyandu.

The limitation of this study is that we could not explore in more detail the obstacles in the data input of the app by cadres because it was not connected with their identification (id) number. While we were advocating with the regency government for bridging the local top-down policy implementation, we had difficulties getting permission for the top-down approaches in the national government regarding the back-end integration between the iPosyandu app and national governmental apps.

## 5. Conclusions

The data quality based on the dimensions (completeness, accuracy, and consistency) of the secondary data collected from the app database in the last three years showed that some of the data were of good quality. This research also identified the obstacles and offered suggestions for improving the implementation of iPosyandu. A bridge between bottom-up and top-down approaches is needed to implement the policy. The health informatics tools bridge the community and the government in recording and reporting the results of *Posyandu* activities. Advocacy steps are necessary to maximize the HIT benefit in the bridging approaches, which is expected to produce more efficient policy implementation.

## 6. Patents

The iPosyandu application copyright has been registered since 2018 with the number 000103655 in Indonesia's Ministry of Law and Human Rights.

**Author Contributions:** Conceptualization, A.F., F.R.R. and K.M.; methodology, A.F., F.R.R. and K.M.; software, A.F., F.R.R. and A.I.S.; validation, A.F., F.R.R., K.M., A.I.S. and D.F.; formal analysis, A.F. and F.R.R.; investigation, A.F., F.R.R., W.G.P., D.F., A.I.S., D., N.I. and S.N.F.; resources, A.F., F.R.R. and W.G.P.; data curation, A.F., F.R.R., K.M. and D.F.; writing—original draft preparation, A.F., F.R.R., K.M. and D.F.; writing—review and editing, A.F. and F.R.R.; visualization, A.F., F.R.R., W.G.P. and K.M.; supervision, F.R.R., K.M. and D.F.; project administration, A.F., A.I.S. and D.; funding acquisition, A.F., F.R.R., D.F., A.I.S. and D. All authors have read and agreed to the published version of the manuscript.

**Funding:** This research was funded by the Lecturer Competence Internal Grant of Universitas Padjadjaran, Indonesia, grant number 2476/UN6.C/LT/2018; the Indonesia Endowment Fund for Education abbreviated LPDP (*Lembaga Pengelola Dana Pendidikan*); the Ministry of Finance, grant number PRJ-70/LPDP/2019; Telkom Education Foundation (*Yayasan Pendidikan Telkom*) number 0032.1/00/DGSHK02/YPT/2019; and the Kreasi Insani Persada Foundation, grant number (-).

**Institutional Review Board Statement:** The study was conducted in accordance with the Declaration of Helsinki, and approved and registered at Komisi Etik Penelitian (Research Ethics Commission) Universitas Padjadjaran, Number: 160/UN6.KEP/EC/2022.

**Informed Consent Statement:** Informed consent was obtained from all subjects involved in the study.

**Data Availability Statement:** Not applicable.

**Acknowledgments:** The authors acknowledge the Purwakarta Regency Office for permission to conduct the app's research and development, including integrating the iPosyandu app into the government health information system. We also acknowledge all Posyandu, community healthcare center/Puskesmas, and regency/city health offices in Indonesia that have used the app to support their work for supporting community health in Indonesia.

**Conflicts of Interest:** The authors declare no conflict of interest. The funders had no role in the design of the study; in the collection, analyses, or interpretation of data; in the writing of the manuscript; or in the decision to publish the results.

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
