# Peer review of "Posyandu Application in Indonesia: From Health Informatics Data Quality Bridging Bottom-Up and Top-Down Policy Implementation"

_informatics, doi:10.3390/informatics9040074_

Round 1

Reviewer 1 Report

The authors provided clear background information. The research question contributes significantly to the area of public health in Indonesia. With the use of NVivo , the qualitative portion of the research adequately explained the design and results of the study. However, the quantitative design and quantitative analysis weren't clear. It is recommended that this part need sufficient additional details to explain how quantitative data was used and analyzed, how it was linked to the results. 

Overall, I would recommend this study after a minor revision in the quantitative section and minor English language revision. It's an interesting research!

Author Response

September 16, 2022

[Informatics]

Good day,

Please find enclosed our revised version manuscript entitled: Posyandu Application for Children Under-Five: From Health Informatics Data Quality to Bridge between Bottom-Up and Top-Down Policy Implementation in Indonesia.

We responded to the reviewer’s feedback in the attachment. Please see the attachment.

We look forward to hearing from you at your earliest convenience.

Sincerely,

Afina Faza

Master of Public Health Study Program, Faculty of Medicine, Universitas Padjadjaran, Jalan Eyckman No. 38 Gedung RSP Unpad Lantai 4, Bandung 40161, Indonesia;

Biomedical Engineering Study Program, School of Electrical Engineering, Telkom University, Jl. Telekomunikasi No. 1, Terusan Buahbatu—Bojongsoang, Sukapura, Dayeuhkolot, Bandung 40257, Indonesia

Reviewer 2 Report

Dear authors,

Congratulations on a very interesting paper. I personally believe that this paper is insightful with significant information in understanding this field of study. However, I think that the paper needs some minor improvements:

·      Title:

o   The title, although clearly descriptive and clarifying about the content of the text, seems rather long (about 20 words). I would advise shortening it to 12 words maximum.

·      Abstract:

o   At the beginning of the summary, you use an acronym (MCH), what does it stand for? Clarify.

o   I would recommend rewriting the first three sentences to better clarify the reasons for the development of the present research. It would also be interesting to include a short sentence supporting the importance of developing this study.

o   All this would help to create an overall picture of the article from the start.

·      Introduction: Comprehensive, with recent and relevant quotations.

·      Methodology: Use of an experimental design appropriate to the stated research objectives, with clear, obvious and detailed explanations. However:

o   Line 110: How many subjects were finally involved in the qualitative part? What roles/positions/characteristics did they have? I advise you to expand on this. A brief description or an explanatory table would be helpful.

o   Line 112: What is these exclusion/inclusion criteria?

o   I would introduce a short paragraph on the limitations of the study. I would urge to identify the weaknesses of the study as reader also wishes to know what difficulties and challenges for you to conduct such study. 

·      Findings/Results: Clear and detailed. However:

o   It would be helpful, given the organization of this section, if you could add two sub-headings to visually separate quantitative and qualitative results.

·      Conclusions: Complete and referenced with previous literature. However:

o   I would introduce an introductory paragraph briefly summarizing the aim of the research and the methodologies applied.

·      Implications: Fine with them.

·      Tables and figures: They are appropriate and very clear. They are easy to interpret and they support the arguments. However, I would add the source (are they own elaborated, right?).

Finally, congratulations once again to the authors for a very interesting article that, with further improvements, may be considered for publication. I hope my comments will be helpful. 

Sincerely.

Author Response

(The authors gave the same response as above.)

Reviewer 3 Report

Thank you for the article describing data quality issues in implementing health promotion empowerment strategies in Indonesia. There are some comments:
- Please provide the full name that MCH stands for, as Informatics readers may not know it. Not all of them are public health experts.
- Please give us more details about Posyandu and iPosyandu. Could you provide a diagram to the audience to better illustrate Posyandu? Could you also provide a graphical interface of iPosyandu?
- It would be nice if you could relate your analysis to other factors that might affect consistency, completeness, and accuracy. The paper itself is too descriptive.
- Figure 4 has many interesting points that should be highlighted. Would you please describe each issue in more detail?
- I think this study needs a more detailed analysis of the results. The qualitative part is pretty bland.

Author Response

(The authors gave the same response as above.)

Round 2

Reviewer 3 Report

Many thanks to the authors for revising the manuscript.

The authors have addressed all the points that I was concerned about. The paper is now more complete and easier to read. 

In my opinion, the paper can now be accepted with minor changes to some typographical and formatting errors.